# Few-Shot Adversarial Domain Adaptation

**Saeid Motiian, Quinn Jones, Seyed Mehdi Iranmanesh, Gianfranco Doretto**
Lane Department of Computer Science and Electrical Engineering
West Virginia University
`{samotiian, qjones1, seiranmanesh, gidoretto}@mix.wvu.edu`

## Abstract

This work provides a framework for addressing the problem of supervised domain adaptation with deep models. The main idea is to exploit adversarial learning to learn an embedded subspace that simultaneously maximizes the confusion between two domains while semantically aligning their embedding. The supervised setting becomes attractive especially when there are only a few target data samples that need to be labeled. In this *few-shot learning* scenario, alignment and separation of semantic probability distributions is difficult because of the lack of data. We found that by carefully designing a training scheme whereby the typical binary adversarial discriminator is augmented to distinguish between four different classes, it is possible to effectively address the supervised adaptation problem. In addition, the approach has a high "speed" of adaptation, i.e. it requires an extremely low number of labeled target training samples, even one per category can be effective. We then extensively compare this approach to the state of the art in domain adaptation in two experiments: one using datasets for handwritten digit recognition, and one using datasets for visual object recognition.

## 1   Introduction

As deep learning approaches have gained prominence in computer vision we have seen tasks that have large amounts of available labeled data flourish with improved results. There are still many problems worth solving where labeled data on an equally large scale is too expensive to collect, annotate, or both, and by extension a straightforward deep learning approach would not be feasible. Typically, in such a scenario, practitioners will train or reuse a model from a closely related dataset with a large amount of samples, here called the source domain, and then train with the much smaller dataset of interest, referred to as the target domain. This process is well-known under the name finetuning. Finetuning, while simple to implement, has been found to be sub-optimal when compared to later techniques such as *domain adaptation* [5]. Domain Adaptation can be *supervised* [58, 27], *unsupervised* [15, 34], or *semi-supervised* [16, 21, 63], depending on what data is available in a labeled format and how much can be collected.

Unsupervised domain adaptation (UDA) algorithms do not need any target data labels, but they require large amounts of target training samples, which may not always be available. Conversely, supervised domain adaptation (SDA) algorithms do require labeled target data, and because labeling information is available, for the same quantity of target data, SDA outperforms UDA [38]. Therefore, if the available target data is scarce, SDA becomes attractive, even if the labeling process is expensive, because only few samples need to be processed.

Most domain adaptation approaches try to find a feature space such that the confusion between source and target distributions in that space is maximum (*domain confusion*). Because of that, it is hard to say whether a sample in the feature space has come from the source distribution or the target distribution. Recently, generative adversarial networks [18] have been introduced for image generation which can also be used for domain adaptation. In [18], the goal is to learn a discriminator

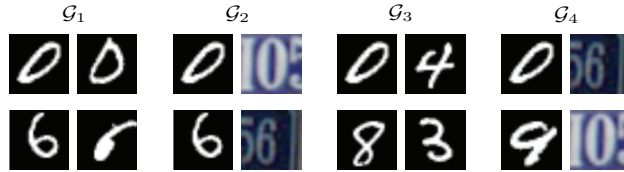

Figure 1: **Examples from MNIST [32] and SVHN [40] of grouped sample pairs.** $\mathcal{G}_1$ is composed of samples of the same class from the source dataset in this case MNIST. $\mathcal{G}_2$ is composed of samples of the same class, but one is from the source dataset and the other is from the target dataset. In $\mathcal{G}_3$ the samples in each pair are from the source dataset but with differing class labels. Finally, pairs in $\mathcal{G}_4$ are composed of samples from the target and source datasets with differing class labels.

to distinguish between real samples and generated (fake) samples and then to learn a generator which best confuses the discriminator. Domain adaptation can also be seen as a generative adversarial network with one difference, in domain adaptation there is no need to generate samples, instead, the generator network is replaced with an inference network. Since the discriminator cannot determine if a sample is from the source or the target distribution the inference becomes optimal in terms of creating a joint latent space. In this manner, generative adversarial learning has been successfully modified for UDA [33, 59, 49] and provided very promising results.

Here instead, we are interested in adapting adversarial learning for SDA which we are calling *few-shot adversarial domain adaptation (FADA)* for cases when there are very few labeled target samples available in training. In this *few-shot learning* regime, our SDA method has proven capable of increasing a model's performance at a very high rate with respect to the inclusion of additional samples. Indeed, even one additional sample can significantly increase performance.

Our first contribution is to handle this scarce data while providing effective training. Our second contribution is to extend adversarial learning [18] to exploit the label information of target samples. We propose a novel way of creating pairs of samples using source and target samples to address the first challenge. We assign a group label to a pair according to the following procedure: 0 if samples of a pair come from the source distribution and the same class label, 1 if they come from the source and target distributions but the same class label, 2 if they come from the source distribution but different class labels, and 3 if they come from the source and target distributions and have different class labels. The second challenge is addressed by using adversarial learning [18] to train a deep inference function, which confuses a well-trained domain-class discriminator (DCD) while maintaining a high classification accuracy for the source samples. The DCD is a multi-class classifier that takes pairs of samples as input and classifies them into the above four groups. Confusing the DCD will encourage *domain confusion*, as well as the *semantic alignment* of classes. Our third contribution is an extensive validation of FADA against the state-of-the-art. Although our method is general, and can be used for all domain adaptation applications, we focus on visual recognition.

## 2 Related work

Naively training a classifier on one dataset for testing on another is known to produce sub-optimal results, because an effect known as dataset bias [42, 57, 56] or covariate shift [51] occurs due to a difference in the distributions of the images between the datasets.

Prior work in domain adaptation has minimized this shift largely in three ways. Some try to find a function which can map from the source domain to the target domain [47, 28, 19, 16, 11, 55, 52]. Others find a shared latent space that both domains can be mapped to before classification [35, 2, 39, 13, 14, 41, 37, 38]. Finally, some use regularization to improve the fit on the target domain [4, 1, 62, 10, 3, 8]. UDA can leverage the first two approaches while SDA uses the second, third, or a combination of the two approaches. In addition to these methods, [6, 36, 50] have addressed UDA when an auxiliary data view [31, 37], is available during training, but that is beyond the scope of this work.

For this approach we are focused on finding a shared subspace for both the source and target distributions. Siamese networks [7] work well for subspace learning and have worked very well with deep convolutional neural networks [9, 53, 30, 61]. Siamese networks have also been useful in

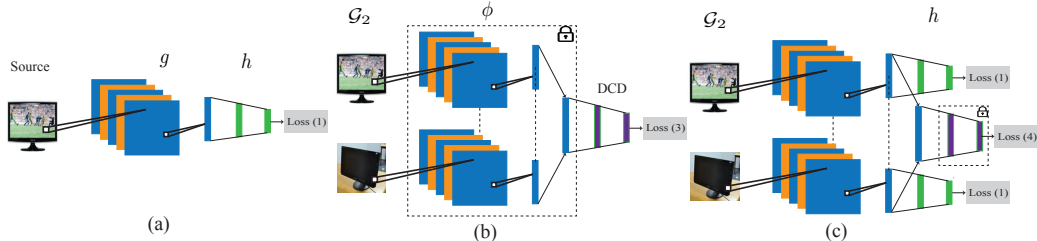

Figure 2: **Few-shot adversarial domain adaptation.** For simplicity we show our networks in the case of weight sharing ($g_s = g_t = g$). **(a)** In the first step, we initialized $g$ and $h$ using the source samples $\mathcal{D}_s$. **(b)** We freeze $g$ and train a DCD. The picture shows a pair from the second group $\mathcal{G}_2$ when the samples come from two different distributions but the same class label. **(c)** We freeze the DCD and update $g$ and $h$.

domain adaptation recently. In [58], which is a deep SDA approach, unlabeled and sparsely labeled target domain data are used to optimize for domain invariance to facilitate domain transfer while using a soft label distribution matching loss. In [54], which is a deep UDA approach, unlabeled target data is used to learn a nonlinear transformation that aligns correlations of layer activations in deep neural networks. Some approaches went beyond Siamese weight-sharing and used couple networks for DA. [27] uses two CNN streams, for source and target, fused at the classifier level. [45], which is a deep UDA approach and can be seen as an SDA after fine-tuning, also uses a two-streams architecture, for source and target, with related but not shared weights. [38], which is an SDA approach, creates positive and negative pairs using source and target data and then finds a shared feature space between source and target by bringing together the positive pairs and pushing apart the negative pairs.

Recently, adversarial learning [18] has shown promising results in domain adaptation and can be seen as examples of the second category. [33] introduced a coupled generative adversarial network (CoGAN) for learning a joint distribution of multi-domain images for different applications including UDA. [59] has used the adversarial loss for discriminative UDA. [49] introduces an approach that leverages unlabeled data to bring the source and target distributions closer by inducing a symbiotic relationship between the learned embedding and a generative adversarial framework.

Here we use adversarial learning to train inference networks such that samples from different distributions are not distinguishable. We consider the task where very few labeled target data are available in training. With this assumption, it is not possible to use the standard adversarial loss used in [33, 59, 49], because the training target data would be insufficient. We address that problem by modifying the usual pairing technique used in many applications such as learning similarity metrics [7, 23, 22]. Our pairing technique encodes domain labels as well as class labels of the training data (source and target samples), producing four groups of pairs. We then introduce a multi-class discriminator with four outputs and design an adversarial learning strategy to find a shared feature space. Our method also encourages the semantic alignment of classes, while other adversarial UDA approaches do not.

## 3 Few-shot adversarial domain adaptation

In this section we describe the model we propose to address supervised domain adaptation (SDA). We are given a training dataset made of pairs $\mathcal{D}_s = \{(x_i^s, y_i^s)\}_{i=1}^N$. The feature $x_i^s \in \mathcal{X}$ is a realization from a random variable $X^s$, and the label $y_i^s \in \mathcal{Y}$ is a realization from a random variable $Y^s$. In addition, we are also given the training data $\mathcal{D}_t = \{(x_i^t, y_i^t)\}_{i=1}^M$, where $x_i^t \in \mathcal{X}$ is a realization from a random variable $X^t$, and the labels $y_i^t \in \mathcal{Y}$. We assume that there is a *covariate shift* [51] between $X^s$ and $X^t$, i.e., there is a difference between the probability distributions $p(X^s)$ and $p(X^t)$. We say that $X^s$ represents the *source domain* and that $X^t$ represents the *target domain*. Under this settings the goal is to learn a prediction function $f : \mathcal{X} \to \mathcal{Y}$ that during testing is going to perform well on data from the target domain.

The problem formulated thus far is typically referred to as *supervised domain adaptation*. In this work we are especially concerned with the version of this problem where only very few target labeled

---
**Algorithm 1** FADA algorithm
---
1: Train $g$ and $h$ on $\mathcal{D}_s$ using (1).
2: Uniformly sample $\mathcal{G}_1, \mathcal{G}_3$ from $\mathcal{D}_s \times \mathcal{D}_s$.
3: Uniformly sample $\mathcal{G}_2, \mathcal{G}_4$ from $\mathcal{D}_s \times \mathcal{D}_t$.
4: Train DCD w.r.t. $g_t = g_s = g$ using (3).
5: **while** not convergent **do**
6:     Update $g$ and $h$ by minimizing (5).
7:     Update DCD by minimizing (3).
8: **end while**
---

samples per class are available. We aim at handling cases where there is only one target labeled sample, and there can even be some classes with no target samples at all.

In absence of covariate shift a visual classifier $f$ is trained by minimizing a *classification loss*

$$\mathcal{L}_C(f) = E[\ell(f(X^s), Y)] , \tag{1}$$

where $E[\cdot]$ denotes statistical expectation and $\ell$ could be any appropriate loss function. When the distributions of $X^s$ and $X^t$ are different, a deep model $f_s$ trained with $\mathcal{D}_s$ will have reduced performance on the target domain. Increasing it would be trivial by simply training a new model $f_t$ with data $\mathcal{D}_t$. However, $\mathcal{D}_t$ is small and deep models require large amounts of labeled data.

In general, $f$ could be modeled by the composition of two functions, i.e., $f = h \circ g$. Here $g : \mathcal{X} \to \mathcal{Z}$ would be an inference from the input space $\mathcal{X}$ to a feature or inference space $\mathcal{Z}$, and $h : \mathcal{Z} \to \mathcal{Y}$ would be a function for predicting from the feature space. With this notation we would have $f_s = h_s \circ g_s$ and $f_t = h_t \circ g_t$, and the SDA problem would be about finding the best approximation for $g_t$ and $h_t$, given the constraints on the available data.

If $g_s$ and $g_t$ are able to embed source and target samples, respectively, to a domain invariant space, it is safe to assume from the feature to the label space that $h_t = h_s = h$. Therefore, domain adaptation paradigms are looking for such inference functions so that they can use the prediction function $h_s$ for target samples.

Traditional unsupervised DA (UDA) paradigms try to align the distributions of the features in the feature space, mapped from the source and the target domains using a metric between distributions, Maximum Mean Discrepancy [20] being a popular one and other metrics like Kullback Leibler divergence [29] and Jensen–Shannon [18] divergence becoming popular when using adversarial learning. Once they are aligned, a classifier function would no longer be able to tell whether a sample is coming from the source or the target domain. Recent UDA paradigms try to find inference functions to satisfy this important goal using adversarial learning. Adversarial training looks for a domain discriminator $D$ that is able to distinguish between samples of source and target distributions. In this case $D$ is a binary classifier trained with the standard cross-entropy loss

$$\mathcal{L}_{adv-D}(X_s, X_t, g_s, g_t) = -E[\log(D(g_s(X^s)))] - E[\log(1 - D(g_t(X^t)))] . \tag{2}$$

Once the discriminator is learned, adversarial learning tries to update the target inference function $g_t$ in order to confuse the discriminator. In other words, the adversarial training is looking for an inference function $g_t$ that is able to map a target sample to a feature space such that the discriminator $D$ will no longer distinguish it from a source sample.

From the above discussion it is clear that in order to perform well, UDA needs to align the distributions effectively in order to be successful. This can happen only if distributions are represented by a sufficiently large dataset. Therefore, UDA approaches are in a position of weakness when we assume $\mathcal{D}_t$ to be small. Moreover, UDA approaches have also another intrinsic limitation; even with perfect confusion alignment, there is no guarantee that samples from different domains but with the same class label will map nearby in the feature space. This lack of *semantic alignment* is a major source of performance reduction.

### 3.1 Handling Scarce Target Data

We are interested in the case where very few labeled target samples (as low as 1 sample per class) are available. We are facing two challenges in this setting. First, since the size of $\mathcal{D}_t$ is small, we need to find a way to augment it. Second, we need to somehow use the label information of $\mathcal{D}_t$. Therefore, we create pairs of samples. In this way, we are able to alleviate the lack of training target samples by

pairing them with each training source sample. In [38], we have shown that creating positive and negative pairs using source and target data is very effective for SDA. Since the method proposed in [38] does not encode the domain information of the samples, it cannot be used in adversarial learning. Here we extend [38] by creating 4 groups of pairs ($\mathcal{G}_i, i = 1, 2, 3, 4$) as follows: we break down the positive pairs into two groups (Groups 1 and 2), where pairs of the first group consist of samples from the source distribution with the same class labels, while pairs of the second group also have the same class label but come from different distributions (one from the source and one from the target distribution). This is important because we can encode both label and domain information of training samples. Similarly, we break down the negative pairs into two groups (Groups 3 and 4), where pairs of the third group consist of samples from the source distribution with different class labels, while pairs of the forth group come from different class labels and different distributions (one from the source and one from the target distributions). See Figure 1. In order to give each group the same amount of members we use all possible pairs from $\mathcal{G}_2$, as it is the smallest, and then uniformly sample from the pairs in $\mathcal{G}_1$, $\mathcal{G}_3$, and $\mathcal{G}_4$ to match the size of $\mathcal{G}_2$. Any reasonable amount of portions between the numbers of the pairs can also be used.

In classical adversarial learning we would at this point learn a domain discriminator, but since we have semantic information to consider as well, we are interested in learning a multi-class discriminator (we call it domain-class discriminator (DCD)) in order to introduce *semantic alignment* of the source and target domains. By expanding the binary classifier to its multiclass equivalent, we can train a classifier that will evaluate which of the 4 groups a given sample pair belongs to. We model the DCD with 2 fully connected layers with a softmax activation in the last layer which we can train with the standard categorical cross-entropy loss

$$\mathcal{L}_{FADA-D} = -E[\sum_{i=1}^{4} y_{\mathcal{G}_i} \log(D(\phi(\mathcal{G}_i)))] \, , \tag{3}$$

where $y_{\mathcal{G}_i}$ is the label of $\mathcal{G}_i$ and $D$ is the DCD function. $\phi$ is a symbolic function that takes a pair as input and outputs the concatenation of the results of the appropriate inference functions. The output of $\phi$ is passed to the DCD (Figure 2).

In the second step, we are interested in updating $g_t$ in order to confuse the DCD in such a way that the DCD can no longer distinguish between groups 1 and 2, and also between groups 3 and 4 using the loss

$$\mathcal{L}_{FADA-g} = -E[y_{\mathcal{G}_1} \log(D(\phi(\mathcal{G}_2))) + y_{\mathcal{G}_3} \log(D(\phi(\mathcal{G}_4)))] \, . \tag{4}$$

(4) is inspired by the non-saturating game [17] and will force the inference function $g_t$ to embed target samples in a space that DCD will no longer be able to distinguish between them.

**Connection with multi-class discriminators:** Consider an image generation task where training samples come from $k$ classes. Learning the image generator can be done by any standard $k$-class classifier and adding generated samples as a new class (generated class) and correspondingly increasing the dimension of the classifier output from $k$ to $k + 1$. During the adversarial learning, only the generated class is confused. This has proven effective for image generation [48] and other tasks. However, this is different than the proposed DCD, where group 1 is confused with 2, and group 3 is confused with 4. Inspired by [48], we are able to create a $k + 4$ classifier to also guarantee a high classification accuracy. Therefore, we suggest that (4) needs to be minimized together with the main classifier loss

$$\mathcal{L}_{FADA-g} = -\gamma E[y_{\mathcal{G}_1} \log(D(g(\mathcal{G}_2))) + y_{\mathcal{G}_3} \log(D(g(\mathcal{G}_4)))] + E[\ell(f(X^s), Y)] + E[\ell(f(X^t), Y)] \, , \tag{5}$$

where $\gamma$ strikes the balance between classification and confusion. Misclassifying pairs from group 2 as group 1 and likewise for groups 4 and 3, means that the DCD is no longer able to distinguish positive or negative pairs of different distributions from positive or negative pairs of the source distribution, while the classifier is still able to discriminate positive pairs from negative pairs. This simultaneously satisfies the two main goals of SDA, domain confusion and class separability in the

Table 1: **MNIST-USPS-SVHN datasets.** Classification accuracy for domain adaptation over the MNIST, USPS, and SVHN datasets. $\mathcal{M}, \mathcal{U}$, and $\mathcal{S}$ stand for MNIST, USPS, and SVHN domain. LB is our base model without adaptation. FT and FADA stand for fine-tuning and our method, respectively.

| | | Traditional UDA | | | Adversarial UDA | | | | | | | | | | |
|---|---|---|---|---|---|---|---|---|---|---|---|---|---|---|---|
| | LB | [60] | [45] | [15] | [33] | [59] | [49] | SDA | 1 | 2 | 3 | 4 | 5 | 6 | 7 |
| $\mathcal{M} \to \mathcal{U}$ | 65.4 | 47.8 | 60.7 | 91.8 | 91.2 | 89.4 | 92.5 | FT | 82.3 | 84.9 | 85.7 | 86.5 | 87.2 | 88.4 | 88.6 |
| | | | | | | | | [38] | 85.0 | 89.0 | 90.1 | 91.4 | 92.4 | 93.0 | 92.9 |
| | | | | | | | | FADA | 89.1 | 91.3 | 91.9 | 93.3 | 93.4 | 94.0 | **94.4** |
| $\mathcal{U} \to \mathcal{M}$ | 58.6 | 63.1 | 67.3 | 73.7 | 89.1 | 90.1 | 90.8 | FT | 72.6 | 78.2 | 81.9 | 83.1 | 83.4 | 83.6 | 84.0 |
| | | | | | | | | [38] | 78.4 | 82.2 | 85.8 | 86.1 | 88.8 | 89.6 | 89.4 |
| | | | | | | | | FADA | 81.1 | 84.2 | 87.5 | 89.9 | 91.1 | 91.2 | **91.5** |
| $\mathcal{S} \to \mathcal{M}$ | 60.1 | - | - | 82.0 | 76.0 | - | 84.7 | FT | 65.5 | 68.6 | 70.7 | 73.3 | 74.5 | 74.6 | 75.4 |
| | | | | | | | | FADA | 72.8 | 81.8 | 82.6 | 85.1 | 86.1 | 86.8 | **87.2** |
| $\mathcal{M} \to \mathcal{S}$ | 20.3 | - | - | 40.1 | - | - | 36.4 | FT | 29.7 | 31.2 | 36.1 | 36.7 | 38.1 | 38.3 | 39.1 |
| | | | | | | | | FADA | 37.7 | 40.5 | 42.9 | 46.3 | 46.1 | 46.8 | **47.0** |
| $\mathcal{S} \to \mathcal{U}$ | 66.0 | - | - | - | - | - | - | FT | 69.4 | 71.8 | 74.3 | 76.2 | 78.1 | 77.9 | 78.9 |
| | | | | | | | | FADA | 78.3 | 83.2 | 85.2 | 85.7 | 86.2 | 87.1 | **87.5** |
| $\mathcal{U} \to \mathcal{S}$ | 15.3 | - | - | - | - | - | - | FT | 19.9 | 22.2 | 22.8 | 24.6 | 25.4 | 25.4 | 25.6 |
| | | | | | | | | FADA | 27.5 | 29.8 | 34.5 | 36.0 | 37.9 | 41.3 | **42.9** |

feature space. UDA only looks for domain confusion and does not address class separability, because of the lack of labeled target samples.

**Connection with conditional GANs:** Concatenation of outputs of different inferences has been done before in conditional GANs. For example, [43, 44, 64] concatenate the input text to the penultimate layers of the discriminators. [25] concatenates positive and negative pairs before passing them to the discriminator. However, all of them use the vanilla binary discriminator.

**Relationship between $g_s$ and $g_t$:** There is no restriction for $g_s$ and $g_t$ and they can be constrained or unconstrained. An obvious choice of constraint is equality (weight-sharing) which makes the inference functions symmetric. This can be seen as a regularizer and will reduce overfitting [38]. Another approach would be learning an asymmetric inference function [45]. Since we have access to very few target samples, we use weight-sharing ($g_s = g_t = g$).

**Choice of $g_s$, $g_t$, and $h$:** Since we are interested in visual recognition, the inference functions $g_s$ and $g_t$ are modeled by a convolutional neural network (CNN) with some initial convolutional layers, followed by some fully connected layers which are described specifically in the experiments section. In addition, the prediction function $h$ is modeled by fully connected layers with a softmax activation function for the last layer.

**Training Process:** Here we discuss the training process for the weight-sharing regularizer ($g_s = g_t = g$). Once the inference functions $g$ and the prediction function $h$ are chosen, FADA takes the following steps: First, $g$ and $h$ are initialized using the source dataset $\mathcal{D}_s$. Then, the mentioned four groups of pairs should be created using $\mathcal{D}_s$ and $\mathcal{D}_t$. The next step is training DCD using the four groups of pairs. This should be done by freezing $g$. In the next step, the inference function $g$ and prediction function $h$ should be updated in order to confuse DCD and maintain high classification accuracy. This should be done by freezing DCD. See Algorithm 1 and Figure 2. The training process for the non weight-sharing case can be derived similarly.

## 4 Experiments

We present results using the Office dataset [47], the MNIST dataset [32], the USPS dataset [24], and the SVHN dataset [40].

### 4.1 MNIST-USPS-SVHN Datasets

The MNIST ($\mathcal{M}$), USPS ($\mathcal{U}$), and SVHN ($\mathcal{S}$) datasets have recently been used for domain adaptation [12, 45, 59]. They contain images of digits from 0 to 9 in various different environments including in the wild in the case of SVHN [40]. We considered six cross-domain tasks. The first two tasks include $\mathcal{M} \to \mathcal{U}, \mathcal{U} \to \mathcal{M}$, and followed the experimental setting in [12, 45, 33, 59, 49], which involves randomly selecting 2000 images from MNIST and 1800 images from USPS. For the rest of

Table 2: **Office dataset.** Classification accuracy for domain adaptation over the 31 categories of the Office dataset. $\mathcal{A}$, $\mathcal{W}$, and $\mathcal{D}$ stand for Amazon, Webcam, and DSLR domain. LB is our base model without adaptation.

| | | Unsupervised Methods | | | Supervised Methods | | | |
| | LB | [60] | [34] | [15] | [58] | [27] | [38] | FADA |
| --- | --- | --- | --- | --- | --- | --- | --- | --- |
| $\mathcal{A} \rightarrow \mathcal{W}$ | $61.2 \pm 0.9$ | $61.8 \pm 0.4$ | $68.5 \pm 0.4$ | $68.7 \pm 0.3$ | $82.7 \pm 0.8$ | $84.5 \pm 1.7$ | $\mathbf{88.2 \pm 1.0}$ | $88.1 \pm 1.2$ |
| $\mathcal{A} \rightarrow \mathcal{D}$ | $62.3 \pm 0.8$ | $64.4 \pm 0.3$ | $67.0 \pm 0.4$ | $67.1 \pm 0.3$ | $86.1 \pm 1.2$ | $86.3 \pm 0.8$ | $\mathbf{89.0 \pm 1.2}$ | $88.2 \pm 1.0$ |
| $\mathcal{W} \rightarrow \mathcal{A}$ | $51.6 \pm 0.9$ | $52.2 \pm 0.4$ | $53.1 \pm 0.3$ | $54.09 \pm 0.5$ | $65.0 \pm 0.5$ | $65.7 \pm 1.7$ | $\mathbf{72.1 \pm 1.0}$ | $71.1 \pm 0.9$ |
| $\mathcal{W} \rightarrow \mathcal{D}$ | $95.6 \pm 0.7$ | $98.5 \pm 0.4$ | $\mathbf{99.0 \pm 0.2}$ | $\mathbf{99.0 \pm 0.2}$ | $97.6 \pm 0.2$ | $97.5 \pm 0.7$ | $97.6 \pm 0.4$ | $97.5 \pm 0.6$ |
| $\mathcal{D} \rightarrow \mathcal{A}$ | $58.5 \pm 0.8$ | $52.1 \pm 0.8$ | $54.0 \pm 0.4$ | $56.0 \pm 0.5$ | $66.2 \pm 0.3$ | $66.5 \pm 1.0$ | $\mathbf{71.8 \pm 0.5}$ | $68.1 \pm 06$ |
| $\mathcal{D} \rightarrow \mathcal{W}$ | $80.1 \pm 0.6$ | $95.0 \pm 0.5$ | $96.0 \pm 0.3$ | $\mathbf{96.4 \pm 0.3}$ | $95.7 \pm 0.5$ | $95.5 \pm 0.6$ | $\mathbf{96.4 \pm 0.8}$ | $\mathbf{96.4 \pm 0.8}$ |
| *Average* | 68.2 | 70.6 | 72.9 | 73.6 | 82.2 | 82.6 | **85.8** | 84.9 |

the cross-domain tasks, $\mathcal{M} \rightarrow \mathcal{S}, \mathcal{S} \rightarrow \mathcal{M}, \mathcal{U} \rightarrow \mathcal{S}$, and $\mathcal{S} \rightarrow \mathcal{U}$, we used all training samples of the source domain for training and all testing samples of the target domain for testing.

Since [12, 45, 33, 59, 49] introduced unsupervised methods, they used all samples of a target domain as unlabeled data in training. Here instead, we randomly selected $n$ labeled samples per class from target domain data and used them in training. We evaluated our approach for $n$ ranging from 1 to 4 and repeated each experiment 10 times (we only show the mean of the accuracies for this experiment because standard deviation is very small).

Since the images of the USPS dataset have $16 \times 16$ pixels, we resized the images of the MNIST and SVHN datasets to $16 \times 16$ pixels. We assume $g_s$ and $g_t$ share weights ($g = g_s = g_t$) for this experiment. Similar to [32], we used 2 convolutional layers with 6 and 16 filters of $5 \times 5$ kernels followed by max-pooling layers and 2 fully connected layers with size 120 and 84 as the inference function $g$, and one fully connected layer with softmax activation as the prediction function $h$. Also, we used 2 fully connected layers with size 64 and 4 as DCD (4 groups classifier). Training for each stage was done using the Adam Optimizer [26]. We compare our method with 1 SDA method, under the same condition, and 6 recent UDA methods. UDA methods use all target samples in their training stage, while we only use very few labeled target samples per category in training.

Table 1 shows the classification accuracies across a range for the number of target samples available in training ($n = 1, \dots, 7$). FADA works well even when only one target sample per category ($n = 1$) is available in training. We can get comparable accuracies with the state-of-the-art using only 10 labeled target samples (one sample per class $n = 1$) instead of using more than thousands of unlabeled target samples. We also report the lower bound (LB) of our model which corresponds to training the base model using only source samples. Moreover, we report the accuracies obtained by fine-tuning (FT) the base model on available target data and also the recent work presented in [38]. Although Table 1 shows that FT increases the accuracies over LB, it has reduced performance compared to SDA methods.

Figure 3 shows how much improvement can be obtained with respect to the base model. The base model is the lower bound LB. This is simply obtained by training $g$ and $h$ with only the classification loss and source training data; so, no adaptation is performed.

**Weight-Sharing.** As we discussed earlier, weight-sharing can be seen as a regularizer that prevents the target network $g_t$ from overfitting. This is important because $g_t$ can be easily overfitted since target data is scarce. We repeated the experiment for the $\mathcal{U} \rightarrow \mathcal{M}$ with $n = 5$ without sharing weights. This provides an average accuracy of 84.1 over 10 repetitions, which is less than the weight-sharing case.

## 4.2   Office Dataset

The office dataset is a standard benchmark dataset for visual domain adaptation. It contains 31 object classes for three domains: Amazon, Webcam, and DSLR, indicated as $\mathcal{A}$, $\mathcal{W}$, and $\mathcal{D}$, for a total of 4,652 images. The first domain $\mathcal{A}$, consists of images downloaded from online merchants, the second $\mathcal{W}$, consists of low resolution images acquired by webcams, the third $\mathcal{D}$, consists of high resolution images collected with digital SLRs. We consider four domain shifts using the three domains ($\mathcal{A} \rightarrow \mathcal{W}, \mathcal{A} \rightarrow \mathcal{D}, \mathcal{W} \rightarrow \mathcal{A}$, and $\mathcal{D} \rightarrow \mathcal{A}$). Since there is not a considerable domain shift between $\mathcal{W}$ and $\mathcal{D}$, we exclude $\mathcal{W} \rightarrow \mathcal{D}$ and $\mathcal{D} \rightarrow \mathcal{W}$.

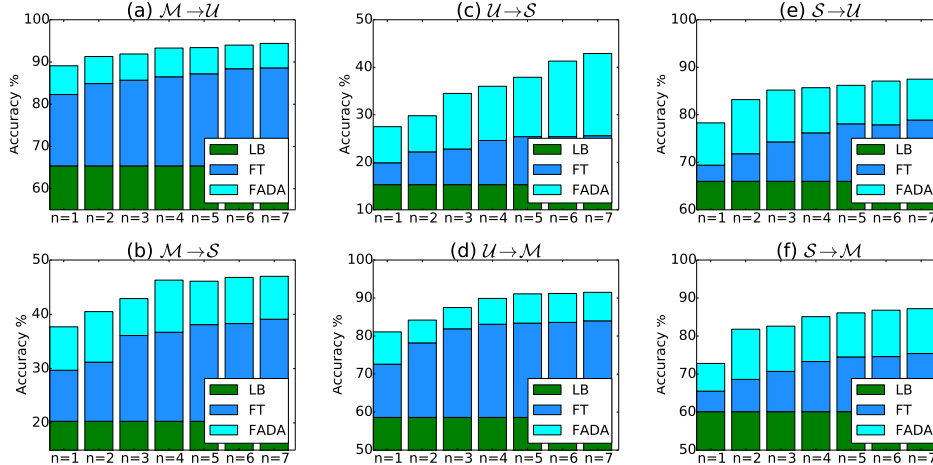

**Figure 3: MNIST-USPS-SVHN summary.** The lower bar of each column represents the `LB` as reported in Table 1 for the corresponding domain pair. The middle bar is the improvement of fine-tuning `FT` the base model using the available target data reported in Table 1. The top bar is the improvement of `FADA` over `FT`, also reported in Table 1.

We followed the setting described in [58]. All classes of the office dataset and 5 train-test splits are considered. For the source domain, 20 examples per category for the Amazon domain, and 8 examples per category for the DSLR and Webcam domains are randomly selected for training for each split. Also, 3 labeled examples are randomly selected for each category in the target domain for training for each split. The rest of the target samples are used for testing. Note that we used the same splits generated by [58].

In addition to the SDA algorithms, we report the results of some recent UDA algorithms. They follow a different experimental protocol compared to the SDA algorithms, and use all samples of the target domain in training as unlabeled data together with all samples of the source domain. So, we cannot make an exact comparison between results. However, since UDA algorithms use all samples of the target domain in training and we use only very few of them (3 per class), we think it is still worth looking at how they differ.

Here we are interested in the case where $g_s$ and $g_t$ share weights ($g_s = g_t = g$). For the inference function $g$, we used the convolutional layers of the VGG-16 architecture [53] followed by 2 fully connected layers with output size of 1024 and 128, respectively. For the prediction function $h$, we used a fully connected layer with softmax activation. Similar to [58], we used the weights pre-trained on the ImageNet dataset [46] for the convolutional layers, and initialized the fully connected layers using all the source domain data. We model the DCD with 2 fully connected layers with a softmax activation in the last layer.

Table 2 reports the classification accuracy over 31 classes for the Office dataset and shows that FADA has performance comparable to the state-of-the-art.

## 5 Conclusions

We have introduced a deep model combining a classification and an adversarial loss to address SDA in few-shot learning regime. We have shown that adversarial learning can be augmented to address SDA. The approach is general in the sense that the architecture sub-components can be changed. We found that addressing the semantic distribution alignments with point-wise surrogates of distribution distances and similarities for SDA works very effectively, even when labeled target samples are very few. In addition, we found the SDA accuracy to converge very quickly as more labeled target samples per category are available. The approach shows clear promise as it sets new state-of-the-art performance in the experiments.

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
