[Reviews · NeurIPS 2017]

Reviewer 1



This paper addressed the problem of supervised adversarial domain adaption with deep models. The application scenario is that there is a small amount of labeled target data. This paper is not well written, and some claims/descriptions are not convincing/clear. Below are detailed comments. (1) Why are there 4 groups (section 3.1) ? More analysis should be given. Given the current description, it is very confusing and the reviewer cannot get the insight why this will work. (2) Lines 114-122: Is it really correct? Source domain and target domain use the same classification function? (3) Experimental justification is not strong: Table 2 shows that proposed approach achieves slight improvement or even the performance is lower than [23].

Reviewer 2



Overall The paper tackles the problem of domain adaptation when a few (up to 4) unlabeled samples exist in the target domain. An approach is presented based on the idea of adversarial training. The main contribution of the paper is in proposing an augmentation approach to alleviate the scarcity of target domain data, that is by taking pairs of source-target samples as input to combinatorially increase the number of available training samples. The approach proves to be very effective on two standard domain adaptation benchmarks, achieving new state-of-the-art results using only 4 labelled samples. The problem is very relevant in general and the achieved results are substantial. This by itself warrants the paper to be published in NIPS. However, several important experiments are missing for a better and in-depth understanding of the proposed method. Should it have included those experiments, I’d have rated it as top 15% papers. Related Works + A structured and thorough coverage of the related works regarding domain adaptation. - related works are missing which have GAN’s discriminator to also predict class labels as in this work - related works are missing on using pairs of samples as for augmenting low number of available data, especially for learning common embedding space Approach + The incremental novelty of the augmentation technique along with the new loss for the context of deep adversarial domain adaptation is a plausible and effective approach. - Many details are missing: batchsize, learning rate, number of general iterations, number of iterations to train DCD and g_t/h, number of trials in adversarial training of the embedding space before convergence, etc. - introduction refers to the case where zero samples from a class is available, however this case is not discussed/studied neither in the approach section nor in the experiments. Experiments + the results using only a few labelled samples are impressive. Specially, when compared with prior and concurrent works using similar approaches. - Why not go above 4? It would be informative to see the diminishing return - A systematic study of the components of the architecture is missing, specially comparing when g_t and g_s are shared and not, but also regarding many other architectural choices. - In sec 3.1 it is mentioned a classification loss is necessary for the better working of the adaptation model. It seems conceptually redundant, except with the added loss more weight can be put on classification as opposed to domain confusion. In that respect, the experimental setup lacks demonstrating the importance of this factor in the loss function. - It seems the most basic configuration of an adversarial training is used in this work, plots and figures showing the stability of convergence using the introduced losses and architectures is important. - An experiment where for some classes no target sample is available?

Reviewer 3



This paper proposed an adversarial framework for the supervised domain adaptation. This framework learns an embedded subspace that maximizes the confusion between two domains while aligning embedded versions through adversarial learning. To deal with the lack of data in target domain, the authors proposed an augmentation method by creating pairs in source and target domains to make the framework work with low number of labeled target samples. There are some problems: 1. The idea is quite simple and intuitive as data augmentation has been widely used in deep learning for small sample problems. As expected, the results are reasonably good. 2. Figures and tables in this paper should be better organized. The figure and corresponding description is far away with each other. For example, figure1 is illustrated in page2. But it is first mentioned in page 4. 3. The pipeline of framework in figure2 is inconsistency with the paper. For example, Φ is a function that concatenates the results of g_t or g_s. But in figure 2(b), Φ also includes the convolution operation. As mentioned in algorithm 1, the g_t and h should be updated by minimizing (5). However, in figure 2(c), it is updated using data in source domain not target domain. 4. To make the experiments consistent, the authors should also run the experiments for several times and give the average accuracy and variation on MNIST-USPS-SVHN datasets. The authors should pay attention to some details in the paper. For example, in figure 4, the vertical axis descriptions on figure4(a)(c) are inconsistent with figure4(b)(d). The horizontal axis descriptions on figure 4(b)(d) is not clear.